# Determination of Electrical and Mechanical Properties of Liquids Using a Resonator with a Longitudinal Electric Field

**DOI:** 10.3390/s24030793

**Published:** 2024-01-25

**Authors:** Alexander Semyonov, Boris Zaitsev, Andrey Teplykh, Irina Borodina

**Affiliations:** Kotel’nikov Institute of Radio Engineering and Electronics of RAS, Saratov Branch, 410019 Saratov, Russia; alex-sheih@yandex.ru (A.S.); teplykhaa@mail.ru (A.T.);

**Keywords:** piezoelectric resonator with a longitudinal electric field, electrical impedance and admittance of the resonator, resonant frequency, conductivity, dielectric constant, viscosity coefficient and elastic module of the liquid

## Abstract

The possibility of determining the elastic modules, viscosity coefficients, dielectric constant and electrical conductivity of a viscous conducting liquid using a piezoelectric resonator with a longitudinal electric field is shown. For the research, we chose a piezoelectric resonator made on an AT-cut quartz plate with round electrodes, operating with a shear acoustic mode at a frequency of about 4.4 MHz. The resonator was fixed to the bottom of a 30 mL liquid container. The samples of a mixture of glycerol and water with different viscosity and conductivity were used as test liquids. First, the frequency dependences of the real and imaginary parts of the electrical impedance of a free resonator were measured and, using the Mason electromechanical circuit, the elastic module, viscosity coefficient, piezoelectric constant and dielectric constant of the resonator material (quartz) were determined. Then, the container was filled with the test sample of a liquid mixture so that the resonator was completely covered with liquid, and the measurement of the frequency dependences of the real and imaginary parts of the electrical impedance of the loaded resonator was repeated. The dependences of the frequency of parallel and series resonances, as well as the maximum values of the electrical impedance and admittance on the conductivity of liquids for various viscosity values, were plotted. It was shown that these dependences can be used to unambiguously determine the viscosity and conductivity of the test liquid. Next, by fitting the theoretical frequency dependences of the real and imaginary parts of the electrical impedance of the resonator loaded with the liquid under study to the experimental dependences, the elastic module of the liquid and its dielectric constant were determined.

## 1. Introduction

Over the years, researchers have been developing and improving acoustoelectric sensors to measure various properties of liquids. The urgency of the problem is related to the need for the quick and accurate analysis of the properties of liquids in such areas as biology, medicine, the food industry, the state of reservoirs and wastewater [1,2].

Particular attention in this regard was paid to acoustic waves propagating in piezoelectric plates. It has been established that the velocity and attenuation of acoustic waves in piezoelectric plates of zero and higher order strongly depend on the viscosity and conductivity of the contacting liquid [3,4,5,6,7,8,9], and liquid sensors can be developed on this basis. The traditional approach is that the higher-order modes are experimentally divided into two groups [5,6,7,8,9]. The first group includes the acoustic modes that are strongly sensitive only to the mechanical properties of the liquid. The modes of the second group respond only to the liquid electrical properties. This approach allows us to simultaneously measure the viscosity and conductivity of a liquid using one channel. There is also a known sensor consisting of three channels with acoustic waves of various types located on the same piezoelectric crystal [9]. This sensor was implemented on a standard wafer of 128-Y-X LiNbO_3_. The test liquid was in contact with the upper surface, and three pairs of interdigital transducers (IDTs) were located on the lower surface. The first pair of IDTs excited and received a surface acoustic wave, which was localized near the bottom surface. Because this wave did not respond to the properties of liquids and depended on temperature, it was used to measure the temperature of the structure. The second pair of IDTs generated an acoustic Lamb wave, which propagated in a completely metalized channel contacting with the test liquid. Thus, this wave did not respond to the conductivity of the liquid but responded to its elastic module and viscosity. Finally, the third pair of IDTs excited a Lamb wave of another type, the propagation path of which was only partially metalized. This wave was chosen specifically due to the high sensitivity to the fluid conductivity and low sensitivity to its viscosity. The measured parameters were the phase and insertion loss of the output signals. The disadvantage of such a sensor, in addition to the complexity of its technical implementation, is the complication of recording and analyzing responses from three channels. This approach is also characterized by low accuracy in measuring the conductivity and viscosity of the liquid since the corresponding responses also depend on the unknown dielectric constant of the liquid and an unknown elastic module.

There are also papers devoted to the development of liquid sensors based on surface acoustic waves with the shear horizontal polarization (SH-SAW) [10,11,12,13,14] propagating in a lithium tantalate plate of the 36-YX cut. Two pairs of interdigital transducers, forming two acoustic channels, were deposited on a single plate. One channel was electrically free, and the second one was electrically shorted. There was a liquid container on the top, covering both channels. The container was filled with the test liquid, and the phase and insertion loss at the output of each channel relative to the reference signal were measured. The signal at the output of the short-circuited channel depended only on the mechanical properties of the liquid (viscosity, density and elastic module). The signal at the output of an electrically free channel was determined by the mechanical and electrical properties of the liquid. Such sensors were successfully used to analyze the quality and identification of liquids such as fruit juices [11], tea [12], motor oil [13] and methanol fuel [14]. However, this approach also does not allow for the separate determination of the elastic module, viscosity coefficient, conductivity and dielectric constant of liquids.

Recently, piezoelectric resonators with a lateral electric field have attracted special attention from the developers of liquid sensors [15,16,17,18,19,20,21]. The electrodes of such a resonator were located on one side of the piezoelectric plate, and an acoustic wave propagated in the space between them, repeatedly reflecting between the sides of the plate. This design was very convenient for developing a liquid analyzer since the container with the liquid and the resonator electrodes for picking up the information signal were located on opposite sides of the piezoelectric plate. In addition, such resonators, unlike the traditional resonators with a longitudinal electric field, were very promising for practical use since the parameters of the resonator depended not only on the changes in the mechanical properties of the contacting medium but also on changes in its electrical properties. Such sensors were studied for many years [15,16,17,18], and it was shown that their resonant frequency depended on the conductivity and viscosity of the contacting fluid. However, the change in the resonant frequency did not exceed 1–2% when the viscosity and conductivity of the liquid varied over a wide range. This led to the need to use precision equipment to record small changes in the resonant frequency. In addition, it is obvious that by using only one measured parameter, one cannot determine all the electrical and mechanical parameters of the fluid. An equivalent circuit of a quartz-based lateral electric field resonator was developed, which allowed the estimation of the shift of the resonant frequency with changes in the viscosity, conductivity and dielectric conductivity of the contacting liquid [18]. However, the authors did not use this scheme to determine the mechanical and electrical parameters of the fluid. The liquid sensors based on such resonators were developed [19,20,21] for measuring the changes in the viscosity and electrical conductivity of the contacting liquid, as well as the changes in the dielectric constant. It was also found that a more convenient informative parameter was the impedance or admittance modulus at a fixed frequency near the frequency of parallel or series resonances [20]. In this case, the specified parameter changed by 30–70%, which provided higher sensitivity to changes in the viscosity and conductivity of the liquid. Currently, there is only one known paper [22] that shows the possibility of separately determining the mechanical and electrical properties of liquids based on a resonator with a lateral electric field. The method for the simultaneous determination of the module of elasticity, viscosity coefficient and permittivity of a liquid was based on the use of Mason’s electromechanical equivalent scheme.

There is also the possibility of determining fluid parameters using resonators with a longitudinal electric field. A resonator with a longitudinal electric field based on an AT-cut quartz plate, one side of which was in contact with a viscous liquid, was studied theoretically and experimentally in [23,24]. An analytical expression was obtained that related the parallel resonance frequency shift to the viscosity and density of liquid and quartz, assuming the absence of electrodes. The calculation results obtained for the aqueous solutions of glucose, sucrose and ethanol turned out to be in good agreement with the experimental data. The influence of various liquid parameters on the parallel resonance frequency of a quartz resonator completely immersed in a fluid was also studied in [25]. It was shown that a change in the frequency of such a resonator with a change in the temperature of the liquid was associated with a change in its viscosity and density. It was also established that immersing the resonator in conducting salt solutions led to an increase in frequency (partial dissolution of the electrodes) or a decrease in frequency (deposition of the additional metal layers). It was shown theoretically and experimentally that contact with viscose solutions of the sucrose and glycerol reduced the resonant frequency of parallel resonance. It was also shown theoretically and experimentally that changes in the viscosity, conductivity and dielectric constant of a liquid contacting with a quartz resonator led to a change in the resonant frequency [26]. But, it is obvious that by measuring only one parameter of the resonator, one cannot extract information about the above-mentioned parameters of the liquid. The possibility of the resonators excited by a longitudinal electric field for determining the viscosity coefficient and elastic modulus of a liquid was demonstrated in [27]. It was shown that using these resonators, one can determine the longitudinal and shear elastic modules as well as the longitudinal and shear viscosity coefficients of the suspensions based on glycerol and diamond microparticles.

This paper is devoted to studying the possibility of determining the electrical and mechanical parameters of a liquid using a resonator with a longitudinal electric field with the help of the Mason electromechanical circuit.

## 2. Materials and Methods

### 2.1. Preparation of the Mixtures of Liquids and Measurement of Their Parameters

We investigated the mixtures of distilled water and glycerol with different volumetric concentrations of glycerol (*β*) in water: 0, 44, 65 and 75%. Using an SV-10 viscometer (A&D Company, Tokyo, Japan), the values of the shear viscosity coefficient (*η*_66_*^ll^*) of these mixtures were measured, which turned out to be 0.9, 4.1, 17 and 31 mPa s, respectively. Next, a certain amount of sodium chloride was added to each sample of the mixture, followed by stirring on a magnetic stirrer for 1 h. As a result, 20 liquid samples were prepared with the above viscosity values and with the different conductivities (σ*^l^*): 1.4, 27, 55, 82 and 123 μS/cm. The conductivity of the obtained samples was measured using a HI8733 conductivity meter (HANNA Instruments, Woonsocket, RI, USA). The density of the mixture of the obtained samples was also measured by weighing the fixed volume (1 mL) on a Pioneer PA-214C analytical balance (OHAUS Corporation, Parsippany, NJ, USA). The volume of the weighed sample of the mixture was set using a precision laboratory pipette, Lenpipet Color. In addition, the relative dielectric constant *ε^l^* of the listed liquid samples with different viscosity and conductivity was also determined. This was performed using a flat capacitor consisting of two plane-parallel rectangular metal plates with the shear dimensions of 31 × 24 mm^2^, 2 mm thick, with a gap of 2.4 mm between them. Using the contact rods (Figure 1), the capacitor was placed in a cell for the test liquid and connected to an E4990A impedance meter (Keysight Technologies, Santa Rosa, CA, USA), measuring the capacitance and conductivity of the samples.

The cell was filled with the liquid under study, and the capacitance of this capacitor, completely immersed in the liquid, was measured at a frequency of ~4.4 MHz, corresponding to the operating frequency of the resonator. It was assumed that the capacitance of such a capacitor is proportional to the dielectric constant of the liquid. Using the known dielectric constant of water (*ε^l^* = 80), the dielectric constant of all mixtures was determined.

### 2.2. Description of a Quartz Resonator and Methods for Measuring Its Parameters

The selected resonator based on the quartz plate of AT- or YXl/+35°-cut (Euler angles = 0, 0, 35°) for studying liquids had a plate thickness of 370 µm and electrode diameter of 5.8 mm. The electrode material is a complex silver-based alloy. This cut of quartz is characterized by a low-temperature delay coefficient for the shear acoustic mode excited in this resonator. The quality factor of the parallel resonance of the unloaded resonator turned out to be quite high (Q = 1,300,000), which indicates an insignificant roughness of the quartz surface and electrodes, as well as a high degree of plane-parallelism of the faces [28]. To carry out the experiments, the resonator was fixed into the bottom of a 30 mL plastic container (Figure 2).

First, the frequency dependences of the real and imaginary parts of the electrical impedance of the free resonator were measured using an E4990A impedance analyzer (Keysight Technologies, Santa Rosa, CA, USA). Then, the container was filled with the liquid sample under study so that the resonator was completely immersed in the liquid, and the measurement of the above frequency dependences was repeated. Each liquid sample, immediately before measurement, was stirred on a magnetic stirrer for one hour. The temperature in the room with measuring equipment was maintained within 25–27 °C.

### 2.3. The Influence of the Test Liquid on the Characteristics of a Quartz Resonator

The main characteristics of piezoelectric resonators include the values of the resonant frequency of the parallel and series resonances and the maximum values of the real parts of the electrical impedance and admittance [20,27]. It is well known that these characteristics strongly depend on the electrical and mechanical parameters of contacting liquid [16,20,23,24,25,26,27]. Figure 3 shows the dependences of the frequency of the parallel *F_par_* (a) and serial *F_ser_* (b) resonances and the maximum values of the electrical impedance *R_max_* (c) and admittance *G_max_* (d) on the conductivity of the contacting liquid at the various concentration of glycerol.

The dependences presented in Figure 3 show that with an increase in the conductivity of the liquid from 1.4 to 123 μS/cm, the parallel resonance frequency of the quartz resonator monotonically increases for all “water–glycerol” mixtures. This result is in full qualitative agreement with the data of [26], where the theoretical and experimental dependences of the parallel resonance frequency shift on the conductivity of the contacting liquid are presented. Figure 3d shows that the maximum value of the electrical admittance also increases monotonically with increasing conductivity. With an increase in the conductivity of an aqueous solution of sodium chloride (Figure 3c, curve 1), the maximum value of the real part of the electrical impedance decreases, and for mixtures of “water–glycerol” (Figure 3c, curves 2–4), it increases. In this case, the resonant frequency of the series resonance (Figure 3b) depends only on the concentration of glycerol (viscosity) and practically does not change with changes in the conductivity of the liquid [23,24,25].

Figure 3a,b,d allowed plotting the dependences of the frequency of the parallel *F_par_* and serial *F_ser_* resonances and the maximum value of the real part of the electrical admittance *G_max_* on the measured magnitude of the liquid viscosity at different values of its conductivity. These dependences are presented in Figure 4. One can see that *F_par_* monotonically decreases with increasing viscosity, which is in qualitative agreement with the results of [23,24,25,26]. At that, the increase of the viscosity monotonically reduces the value of *G_max_*. Figure 4b shows that with increasing the viscosity, *F_ser_* reduces monotonically and does not depend on conductivity [23,24,25]. Thus, Figure 4b can be considered a calibration curve for determining the viscosity from the measured value of the resonant frequency *F_ser_*. Figure 4a, in turn, allows us to find the conductivity of the liquid from the measured value of the parallel resonance frequency *F_par_* and the found viscosity value.

### 2.4. Determination of Material Constants of a Resonator Using an Equivalent Circuit

For the calculation of the frequency dependencies of the electrical impedance of a piezoelectric resonator without liquid loading, we used the method based on the equivalent electromechanical scheme of Mason [29]. Since this method is described in detail in [27], we provide only brief information here. Using the equivalent scheme of Mason and the pointed above method, the frequency dependences of the real and imaginary parts of the electric impedance of the free resonator were calculated. We used the orthogonal coordinate system corresponding to the AT cut of the quartz: the X_1_ axis is normal to the surface of the plate and determines the direction of the propagation of the wave, and the X_2_ axis is parallel to the vector of polarization of the wave. In this case, the quartz is described by the elastic module *C*_66_, viscosity coefficient *η*_66_, piezoconstant *e*_16_ and permittivity *ε*_11_ in the indicated coordinate system. As for the material of the electrodes, we will assume that it is isotropy, and the elastic module and viscosity coefficient can be represented as *C*_66_^e^ and *η*_66_^e^.

In this calculation, the above-pointed material constants of quartz and electrodes were taken from the literary sources [27]. Then, in order to determine the true material constant of quartz and electrodes, the calculated frequency dependences of the real and imaginary parts of the electrical impedance were fitted to the experimental ones.

By trying all the unknown constants, *C*_66_, *η*_66_, *e*_16_ and *ε*_11_ of quartz, using the method of the smallest squares, such a combination was determined, for which theoretical frequency dependences of the real (*R*) and imaginary (*X*) parts of the resonator’s electrical impedance were most consistent with experimental ones (Figure 5). Table 1 presents the initial (reference) [27] and the values obtained as a result of fitting.

### 2.5. Determination of Material Constants of the Liquids Using an Equivalent Electromechanical Circuit

Figure 6 shows an equivalent electromechanical circuit describing a piezoelectric resonator with a longitudinal electric field completely immersed in a viscous conducting liquid. The mechanical contact of the resonator with a liquid is described by the complex impedance *Z^l^*. The dielectric constant and electrical conductivity of the liquid are taken into account by including in the electrical part of the circuit the additional capacitance *C_a_* and additional resistance *R_a_*, respectively. For a liquid, the mechanical impedance *Z_l_* and the acoustic wave velocity *υ^l^* can be represented as follows [29]:

*Z^l^ = iz^l^S*,(1)

*υ^l^ =* {(*C*_66_*^l^ + iωη*_66_*^l^*)*/ρ^l^*}^1/2^,(2)

*z^l^* = {(*C*_66_*^l^* + *iωη*_66_*^l^*)*ρ^l^*}^1/2^.(3)

Here, *z^l^* is the specific mechanical impedance of the liquid, *C*_66_*^l^* is the shear module of elasticity of the liquid, *η*_66_*^l^* is the shear viscosity coefficient, *ρ^l^* is the density of the liquid, *S* is the area of the electrodes, *i* is the imaginary unit. The subscript *l* indicates that the quantity belongs to the liquid.

According to the method described in [27], such a set of the pointed quantities, *C*_66_*^l^*, *η*_66_*^l^*, *C_a_* and *R_a_*, was determined, for which the theoretical frequency dependences of the real (*R*) and imaginary (*X*) parts of the electrical impedance of the resonator immersed in the liquid corresponded as much as possible to the experimental ones. In the calculations, we used the constants of the piezoelectric material and electrodes obtained earlier in Section 2.4. As an example, Figure 7 shows the theoretical and experimental frequency dependences of the real (*R*) and imaginary (*X*) parts of the electrical impedance of a quartz resonator immersed in an aqueous solution of NaCl with a conductivity of 1.4 μS/cm (Figure 7a,b) and in a “water–glycerol” mixture (*β* = 75%, *σ^l^* = 55 μS/cm) (Figure 7c,d). There is a good agreement between the theory and experiment, and the maximum differences are equal to 6.8% and 2.4% for the real and imaginary parts of the impedance, respectively.

## 3. Results

Table 2 shows the full data on the liquid samples under study. The first four columns indicate the liquid parameters that were directly measured according to the procedures described in 2.1. These are the percentage of glycerol (*β*), dielectric constant (*ε^l^*), density (*ρ^l^*), specific conductivity (*σ^l^*) and shear viscosity measured using a viscometer (*η*_66_*^ll^*). The following columns contain the quantities found using the equivalent circuit of the resonator. These are the shear elastic module (*C*_66_*^l^*), shear viscosity coefficient (*η*_66_*^l^*), shear acoustic wave velocity (*v*_66_*^l^*), additional capacitance (*C_a_*) and additional resistance (*R_a_*), respectively.

From Table 2, one can see that in all cases, the shear viscosity value obtained using Mason’s scheme exceeds the value measured by a viscometer. This is due to the fact that in Mason’s scheme, viscosity describes all the losses that exist in the liquid. These include dielectric and thermoelastic losses, absorption associated with the intrinsic viscosity of the fluid, and acoustic wave scattering associated with the roughness of the resonator surface. The maximum difference in the values of the effective and true viscosity varies from 35% to 6% as the viscosity increases. This is explained by the fact that as viscosity increases, the proportion of the loss associated specifically with the true viscosity of the liquid increases in comparison with other sources of loss. In this case, the spread of viscosity values depending on the conductivity changes from 20% to 3% with increasing the viscosity obtained as a result of fitting the theory to the experiment.

Table 2 also shows that the shear viscosity coefficient *η*_66_*^l^*, the velocity of the shear acoustic wave *υ*_66_*^l^* and the additional capacitance *C_a_* slightly depend on the conductivity of the liquid. In this case, with increasing concentration of glycerol in water, the shear viscosity coefficient *η*_66_*^l^* and the velocity of the shear acoustic wave *υ*_66_*^l^* increase, and the additional capacity *C_a_* decreases. The additional resistance *R_a_* of a resonator immersed in a liquid decreases with increasing the liquid conductivity for all four types of liquids.

Based on the data presented in Table 2, the dependences of the additional capacity *C_a_* on the conductivity of the liquid were constructed for various values of the volume concentration of glycerol. They are shown in Figure 8a. It can be seen that this capacity is practically independent of the conductivity of the liquid and decreases with increasing the glycerol concentration. The additional capacity *C_a_* is an addition to the capacitance of the free resonator *C*_0_ due to the permittivity of the contacting liquid. With increasing the glycerol concentration, the permittivity of the mixture decreases from 81 to 57, and this leads to a decrease in the capacity C_a_. This is confirmed in Figure 8b, which shows the dependence of the average value of additional capacity <*C_a_*> on the measured relative dielectric constant *ε^l^*. Averaging was carried out for all conductivity values for each glycerol concentration. One can see that this capacity increases monotonically with increasing liquid permittivity. Therefore, the dependence presented in Figure 8b may be used as a calibration curve to determine the liquid permittivity from the calculated value of the additional capacity.

It is obvious that when the resonator is completely immersed in liquid, the resonator electrodes are shunted by both an additional capacitance *C_a_* (due to the dielectric constant) and an additional active resistance *R_a_* (due to its conductivity). In the equivalent circuit, these elements are connected to the electrical input parallel to the resonator capacitance *C*_0_. Using the data given in Table 2, the dependence of the additional resistance *R_a_* on the conductivity *σ^l^* of the liquid was constructed. This dependence is shown in Figure 9. It can be seen that this resistance is practically independent of the volumetric content of glycerol (that is, on viscosity and permittivity) and monotonically decreases with increasing conductivity of the liquid. Therefore, this dependence can be used as a calibration curve to determine the conductivity *σ^l^* of a liquid from the calculated value *R_a_*.

## 4. Discussion

The measured frequency dependences of the real and imaginary parts of the electrical impedance of the quartz resonator completely immersed in the liquid allowed us to obtain the frequencies of parallel and serial resonances and the maximum values of the electrical impedance and admittance as functions of the liquid conductivity with different viscosities. These data allow us to find the values of the viscosity and conductivity of the liquid. To determine other material constants, such as shear elastic module and permittivity with the help of a quartz resonator with a longitudinal electric field, we used measured frequency dependences of the real and imaginary parts of the electrical impedance of a free resonator and a resonator completely immersed in the liquid under study. For this, an equivalent electromechanical circuit of Mason was used. For the equivalent circuit for a resonator completely immersed in a liquid, in addition to the electromechanical transformer and the standard electrical and mechanical parts of the circuit, we also included liquid loading in the mechanical part and additional capacitance and additional resistance in the electrical part. The additional capacitance and additional resistance were included due to the permittivity and conductivity of the liquid. The material constants and also additional capacitances and resistances were determined by fitting the theoretical frequency dependences of the real and imaginary parts of the electrical impedance of the resonator, calculated using an equivalent circuit, to the experimentally measured ones. As a result, the shear elastic module, viscosity coefficient, and relative dielectric constant, as well as values of the additional capacitance and additional resistance of the resonator loaded by liquids with different viscosities and conductivities, were obtained. An increase in the concentration of glycerol in water leads to an increase in the shear viscosity coefficient and the velocity of the shear acoustic wave and to a decrease in the additional capacity. Table 2 shows that the additional capacitance of a resonator immersed in a conducting liquid weakly depends on the conductivity of the liquid. The value of this quantity, averaged over all conductivity values for each type of liquid, increases with increasing experimentally determined relative dielectric constant. The amount of additional resistance decreases with increasing liquid conductivity in all cases.

## 5. Conclusions

Thus, the results obtained allow us to develop an algorithm for measuring the electrical and mechanical parameters of a liquid using a piezoelectric resonator with a longitudinal electric field. The measured dependences of the frequencies of parallel and serial resonances, as well as the maximum value of the real part of the electrical admittance on the viscosity of the liquid at different values of the conductivity, can be considered calibration curves for a given resonator. Therefore, we can propose the following algorithm for determining fluid parameters. (1) First, the frequency dependences of the real and imaginary parts of the electrical impedance of the free (unloaded with liquid) resonator are measured, and by fitting the theoretical dependences to the experimental ones, the characteristics of all elements of the Mason equivalent circuit are determined. (2) Then, the specified frequency characteristics of the resonator, completely immersed in the test liquid, are measured. (3) Based on the measured values of the resonant frequency of the series resonance, the viscosity of the liquid may be found according to the dependence in Figure 4b. (4) Based on the measured value of the parallel resonance frequency and liquid viscosity using the calibration curve in Figure 4a, the conductivity of the liquid is defined. (5) By fitting the theoretical frequency dependences of the real and imaginary parts of the electrical impedance of the loaded resonator to the experimental ones, the elastic module and permittivity of the liquid are determined.

## Figures and Tables

**Figure 1 sensors-24-00793-f001:**
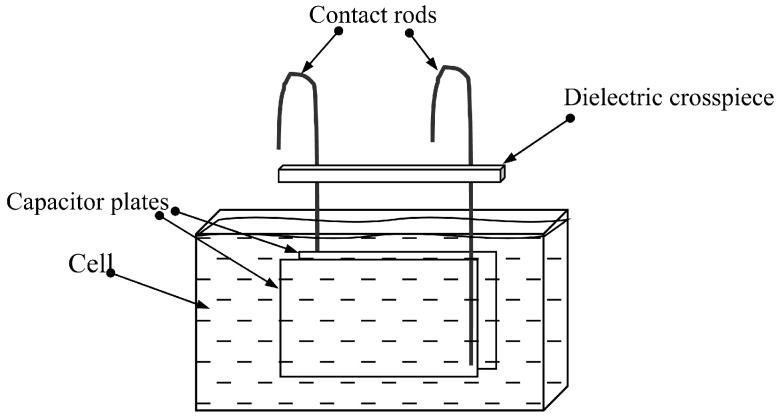
Set-up for determining the dielectric constant of liquids.

**Figure 2 sensors-24-00793-f002:**
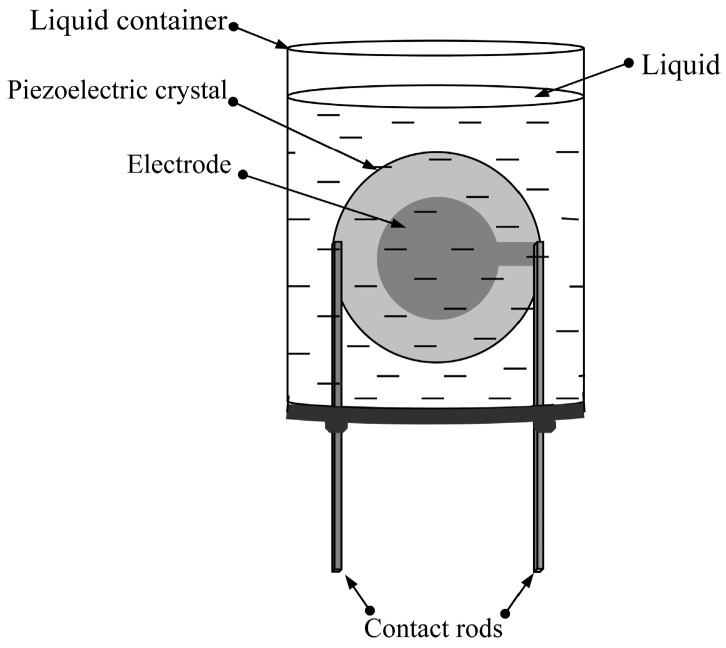
The liquid container with the quartz resonator.

**Figure 3 sensors-24-00793-f003:**
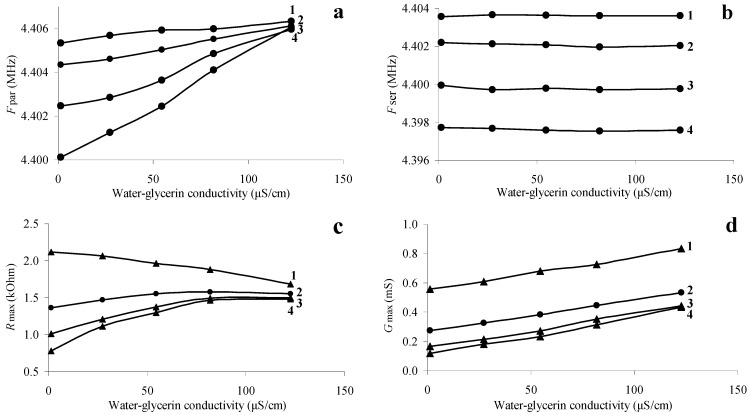
Dependences of the resonant frequency of parallel (**a**) and series (**b**) resonances, as well as the maximum values of the real parts of the electrical impedance (**c**) and admittance (**d**) of a quartz resonator on the conductivity of the liquid. (1—aqueous solution of sodium chloride (*β* = 0), 2—mixture “water-glycerol” *β* = 44%, 3—mixture “water-glycerol” *β* = 65%, 4—mixture “water-glycerol” *β* = 75%).

**Figure 4 sensors-24-00793-f004:**
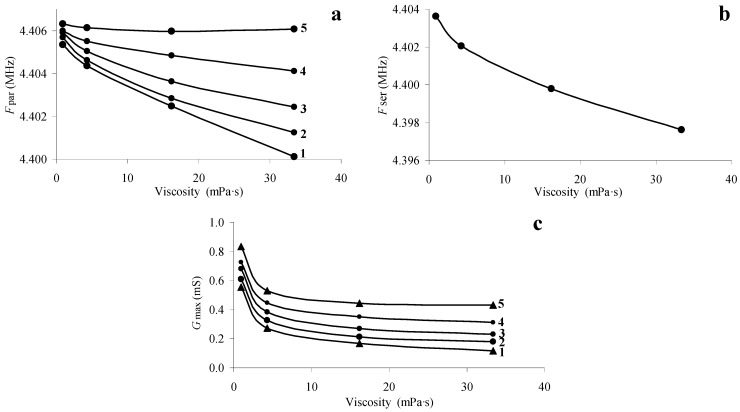
Dependences of the resonant frequency of parallel (**a**) and series (**b**) resonances, as well as the maximum values of the real parts of the electrical admittance (**c**) of a quartz resonator on the viscosity of the liquid. Liquid conductivity: 1—1.4 μS/cm, 2—27 μS/cm, 3—55 μS/cm, 4—82 μS/cm, 5—123 μS/cm.

**Figure 5 sensors-24-00793-f005:**
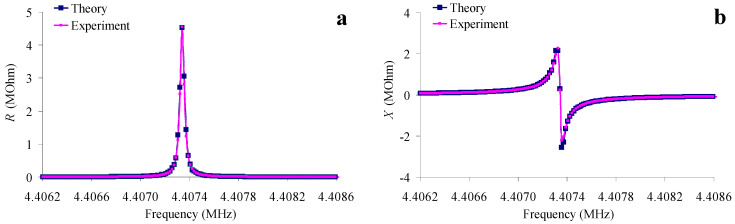
The frequency dependences of the real (**a**) and imaginary (**b**) parts of the electrical impedance of the AT-quartz resonator without load (pink—experiment, blue—the result of fitting).

**Figure 6 sensors-24-00793-f006:**
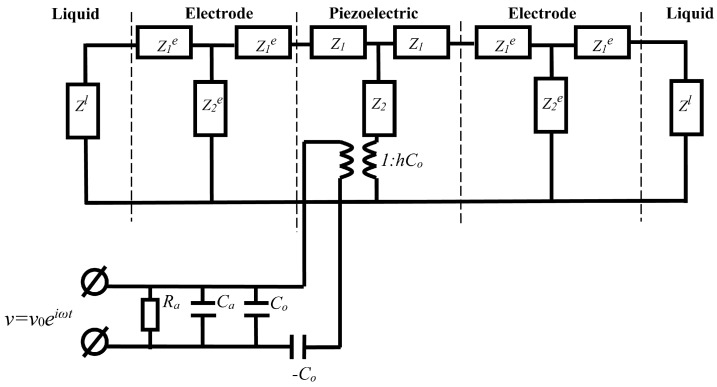
Equivalent circuit of a resonator with electrodes immersed in a liquid, taking into account additional capacitance (*C_a_*) and resistance (*R_a_*).

**Figure 7 sensors-24-00793-f007:**
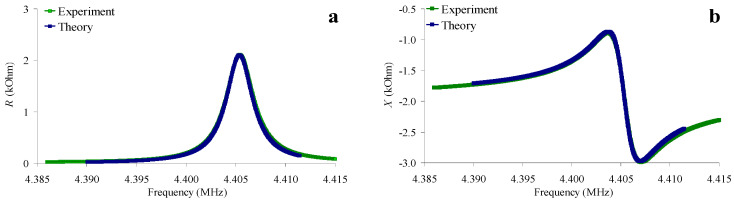
Frequency dependences of the real (**a**,**c**) and imaginary (**b**,**d**) parts of the electrical impedance of a quartz resonator for an aqueous solution of NaCl with a conductivity of 1.4 μS/cm (**a**,**b**), and for a “water–glycerol” mixture *β* = 75%, *σ^l^* = 55 μS/cm (**c**,**d**). Blue color—theory, green color—experiment.

**Figure 8 sensors-24-00793-f008:**
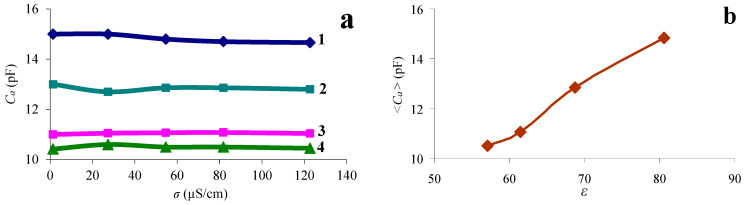
(**a**)—dependence of the additional capacity *C_a_* on the conductivity of the liquid with different percentages of glycerol: 1—water, 2—mixture “water–glycerol” *β* = 44%, 3—mixture “water–glycerol” *β* = 65%, 4—mixture “water–glycerol” *β* = 75%. (**b**)—dependence of the average value of the additional capacity <*Ca*> on the measured value of *ε^l^*.

**Figure 9 sensors-24-00793-f009:**
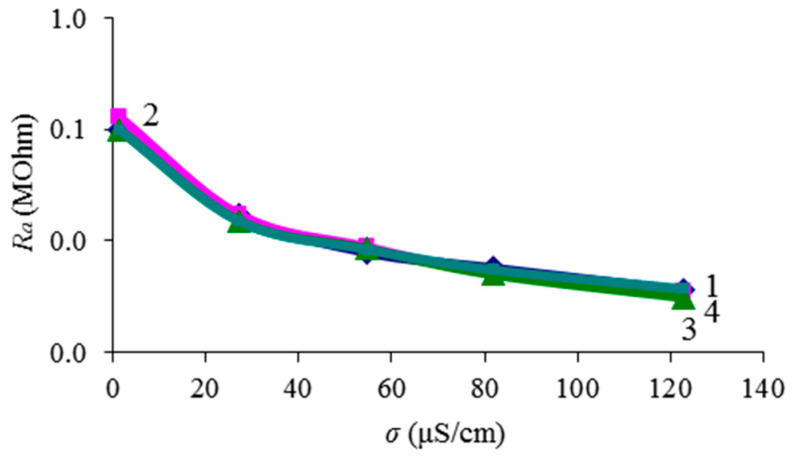
Dependence of the additional resistance *R_a_* on the conductivity of a liquid with different percentages of glycerol: 1—water, 2—water–glycerol mixture *β* = 44%, 3—water–glycerol mixture *β* = 65%, 4—water–glycerol mixture *β* = 75%.

**Table 1 sensors-24-00793-t001:** Initial (reference) and resulting from the fitting values of the material constants of the quartz resonator AT-cut.

	*ρ*, kg/m^3^	*C*_66_, 10^10^ Pa	*η*_66_, Pa·s	*e*_16_, C/m^2^	*ε*_11_, 10^−11^ F/m
The initial values of the material constants of quartz	2648.38	2.899	0	0.067	3.92
The obtained values of the material constants of quartz	2649	2.9	0.009	0.07	4.05
The obtained values of the constants of the electrodes	6700	2.06	0.1		

Data for the material constants of the electrodes are also presented at the bottom line of the table.

**Table 2 sensors-24-00793-t002:** The parameters of liquid samples under study: the percentage of glycerol (*β*), dielectric constant (*ε^l^*), density (*ρ^l^*), specific conductivity (*σ^l^*), shear viscosity measured by a viscometer (*η*_66_*^ll^*), shear elastic module (*C*_66_*^l^*), shear coefficient of viscosity (*η*_66_*^l^*), shear acoustic wave velocity (*v*_66_*^l^*), additional capacitance (*C_a_*) and resistance (*R_a_*).

*β*, %	*ε^l^*	*ρ^l^*, kg/m^3^	*σ^l^*, μS/cm	*η*_66_*^l1^*, mPa·s	*η*_66_*^l^*, mPa·s	*C*_66_*^l^*, 10^4^ Pa	*υ*_66_*^l^*, m/s 4.4 MHz	*C_a_*, pF	*R_a_*, MOhm
0	81	1000	1.36	0.84	1.3	1.6	5.2	15	0.1
			27	0.83	1.0	1.7	5.0	15	0.017
			55	0.9	0.9	2.0	5.1	14.8	0.0079
			82	0.91	1.0	1.9	5.1	14.7	0.0057
			123	0.94	1.1	1.9	5.2	14.7	0.0036
44	69	1100	1.36	4.1	5.3	1.2	8.5	13	0.1
			27	4.1	5.2	1.3	8.5	12.7	0.015
			55	4.1	5.4	1.5	8.7	12.9	0.0084
			82	4.1	5.4	1.5	8.7	12.9	0.0055
			123	4.1	5.1	1.6	8.5	12.8	0.0037
65	62	1170	1.36	16.6	18.1	0.7	14.7	11	0.13
			27	16.2	18.9	0.8	15.1	11.1	0.017
			55	17	19.1	0.9	15.2	11.1	0.0088
			82	17.5	19.2	2.3	15.4	11.1	0.0051
			123	18	19.5	2.7	15.6	11.04	0.0035
75	57	1190	1.36	31.7	34	0.6	19.9	10.4	0.1
			27	31.1	32	0.6	19.3	10.6	0.015
			55	30.2	32.5	2.2	19.7	10.5	0.0085
			82	32.8	33	2.2	19.8	10.5	0.005
			123	33.8	33.5	2.9	20.0	10.5	0.0031

## Data Availability

Data are contained within the article.

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
