# Peer review of "Determination of Electrical and Mechanical Properties of Liquids Using a Resonator with a Longitudinal Electric Field"

_sensors, 2024, doi:10.3390/s24030793_

Round 1

Reviewer 1 Report

Comments and Suggestions for Authors

Very well-written paper.  The introduction was quite thorough and provided a nice concise summary of all relevant background material.  Although there was no major development of a sensor design (i.e. as discussed in the background section, were a singular sensor design is the goal), the work does show simplistically on an approach to define a singular sensor that could characterize conductivity, viscosity and physical properties of a liquid.  Thus, this paper provides a nice insight into the approach one would take to develop a singular MEMs-like sensor to achieve the given goal. 

Author Response

The authors are grateful to Reviewer # 1 for the high assessment of our article.

Reviewer 2 Report

Comments and Suggestions for Authors

The author needs to explain the surface morphology of the QCM sensor electrode and the sensor's flatness as it will affect the series resonance frequency of the sensor, as explained in many papers.

To check the validity of Figure 4 it is good to mention that research on the interaction between QCM sensors and viscosity has been carried out intensively by Kanazawa & Gordon (https://doi.org/10.1016/S0003-2670(00)82721-X); however, the author did not review this publication or other papers. The QCM sensor's response, the relationship between the frequency change and impedance of the sensor, and the viscosity and density of the liquid on the sensor surface have been well explained in many papers.

Repeated measurement experiments to obtain impedance values need to be carried out because impedance measurements have relatively large uncertainties. The author needs to explain more deeply (Figure 4) the relationship between parallel resonance and conductivity because stray capacitance that appears when a conductive or non-conductive liquid will greatly influence the parallel resonance of the QCM. On the other hand, fluid conductivity will not affect series resonance (as explained in the paper by Kanazawa - Gordon).

Table 2 shows data that still does not explain the meaning of the numbers and the relationship between the first three columns of data and the other columns.

Explanation and further investigation are required to connect the data in Figures 8 and 9. Is there any relationship with the QCM data?

The discussion stated that the additional resonator capacitance is weakly related to conductivity. To which resonator capacitance the liquid conductivity has little effect should be clear.

Author Response

The authors are grateful to reviewer No. 2 for useful comments and appreciation of our work.

The author needs to explain the surface morphology of the QCM sensor electrode and the sensor's flatness as it will affect the series resonance frequency of the sensor, as explained in many papers.

Thank you for your comment. It is known that strong roughness and non-planar parallelism of the resonator faces leads to a decrease in the quality factor of both parallel and series resonances as well as to a change in their resonant frequencies [31]. In our case, the quality factor of the parallel resonance of the unloaded resonator turned out to be quite high (Q = 1300000).This indicates an insignificant roughness of the quartz surface and electrodes, as well as a high degree of plane-parallelism of the faces because for resonator on quartz of AT – cut the maximum Qmax =3200000 for 5 MHz [31].

[31] Vig J. R., Quartz Crystal Resonators and Oscillators For Frequency Control and Timing Applications - A Tutorial, US Army Communications-Electronics Research, Development & Engineering Center Fort Monmouth, NJ, USA, 2004

We have added the following text on the page 4:

The quality factor of the parallel resonance of the unloaded resonator turned out to be quite high Q =1300000, which indicates an insignificant roughness of the quartz surface and electrodes, as well as a high degree of plane-parallelism of the faces [31]

[31] Vig J. R., Quartz Crystal Resonators and Oscillators For Frequency Control and Timing Applications - A Tutorial, US Army Communications-Electronics Research, Development & Engineering Center Fort Monmouth, NJ, USA, 2004

To check the validity of Figure 4 it is good to mention that research on the interaction between QCM sensors and viscosity has been carried out intensively by Kanazawa & Gordon (https://doi.org/10.1016/S0003-2670(00)82721-X); however, the author did not review this publication or other papers. The QCM sensor's response, the relationship between the frequency change and impedance of the sensor, and the viscosity and density of the liquid on the sensor surface have been well explained in many papers.

Thank you for this comment. We have added the following text in the introduction on the page 3

There is also the possibility of determining fluid parameters using the resonators with a longitudinal electric field. A resonator with a longitudinal electric field based on an AT cut quartz plate, one side of which was in contact with a viscous liquid, was studied theoretically and experimentally in [27, 28]. An analytical expression was obtained that related the parallel resonance frequency shift to the viscosity and density of liquid and quartz, assuming the absence of electrodes. The calculation results obtained for the aqueous solutions of glucose, sucrose and ethanol turned out to be in good agreement with the experimental data. The influence of various liquid parameters on the parallel resonance frequency of a quartz resonator completely immersed in a fluid was also studied in [29]. It was shown that a change in the frequency of such a resonator with a change in the temperature of the liquid was associated with a change in its viscosity and density. It was also established that immersing the resonator in conducting salt solutions led to an increase in frequency (partial dissolution of the electrodes) or to a decrease in frequency (deposition of the additional metal layers). It was shown theoretically and experimentally that contact with viscose solutions of the sucrose and glycerol reduced the resonant frequency of parallel resonance.

[27] Kanazava, K.K.; Gordon II, J.G. The oscillation frequency of a quartz resonator in contact with a liquid. Analytica Chimica Acta. 1985, 175, 99-105.

[28] Kanazava, K.K.; Gordon II, J.G. Frequency of a Quartz Microbalance in Contact with Liquid. Anal. Chem. 1985, 57, 1770-1771.

[29] Nomura, T.; Watanabe, M.; West, T.S. Behavior of piezoelectric quartz crystals in solutions with the applications to the determination of iodide. Analytica Chimico Acta. 1985, 175, 107-116.

Repeated measurement experiments to obtain impedance values need to be carried out because impedance measurements have relatively large uncertainties.

We have not understood this comment because the frequency dependences of the real and imaginary parts of the electrical impedance of the free and loaded resonator were measured using an E4990A impedance analyzer (Keysight Technologies, Santa Rosa, CA, USA). This information is pointed on the page 4 of our article. This analyzer has an accuracy of less than 1% in the frequency range 1 – 10 MHz and for the impedance range less than 5 MOhm. The repeated measurements give the same values within the limits of accuracy.

The author needs to explain more deeply (Figure 4) the relationship between parallel resonance and conductivity because stray capacitance that appears when a conductive or non-conductive liquid will greatly influence the parallel resonance of the QCM. On the other hand, fluid conductivity will not affect series resonance (as explained in the paper by Kanazawa - Gordon).

The authors are grateful to the Reviewer for this valuable comment. Figure 3a shows the dependence of the parallel resonance frequency on the conductivity of the contacting liquid at different values of its viscosity. Fig. 4a reflects the same result, only viscosity is plotted along the abscissa axis, and conductivity plays the role of a parameter. Figure 3a shows that at a fixed viscosity value, the frequency increases with increasing conductivity. This may be explained by the fact that finite conductivity leads to the additional losses, which always reduce the resonant frequency. This result is in full qualitative agreement with Fig. 8 of the article [30], where the theoretical and experimental dependences of the parallel resonance frequency shift on the conductivity of the contacting liquid are presented. It is obvious that the additional capacitance resulting from the presence of liquid contributes to the change in the parallel resonance frequency. As for the series resonance frequency, as expected, it does not depend on the conductivity of the liquid, as follows from Fig. 3b, but depends on the viscosity.

[30] Josse, F.; Shana, Z.A.; Radtke, D.E.; Haworth, D.T. Analysis of piezoelectric bulk acoustic wave resonators as detectors in viscous conductive liquids. IEEE Trans. Ultrason., Ferroelectr. Freq. Contr. 1990, 37, 359–368.

We have corrected the text on the page 6:

The dependences presented in Figure 3 show that with an increase in the conductivity of the liquid from 1.4 to 123 μS/cm, the parallel resonance frequency (Fig. 3a) and the maximum value of the electrical admittance (Fig. 3d) of the quartz resonator monotonically increases for all “water – glycerol” mixtures. This result is in full qualitative agreement with the data of [30], where the theoretical and experimental dependences of the parallel resonance frequency shift on the conductivity of the contacting liquid are presented. Figure 3d shows that the maximum value of the electrical admittance also increases monotonically with increasing conductivity.

We have corrected also the text on the page 7:

One can see that Fpar and Gmax monotonically decreases with increasing viscosity and monotonically increase with growth of the conductivity, which is in qualitative agreement with the results of [27–30]. At that, the increase of the viscosity monotonically reduces the value of Gmax.

Table 2 shows data that still does not explain the meaning of the numbers and the relationship between the first three columns of data and the other columns.

Thank you for this comment. We agree that it is advisable to indicate all the information about the values given in Table 2 in one place.Thereforewehaveinsertedthefollowingtextonpage 10.

Table 2 shows the full data on the liquid samples under study. The first four columns indicate the liquid parameters that were directly measured according to the procedures described in 2.1. These are the percentage of glycerol (β), dielectric constant (ϵl), density (ρl), specific conductivity (σl) and viscosity measured by a viscometer (η66ll). The following columns contain the quantities found using the equivalent circuit of the resonator. These are the shear elastic module (C66l), shear viscosity coefficient (η66l), shear acoustic wave velocity (v66l), additional capacitance (Ca) and additional resistance (Ra), respectively.

We have changed the caption for the Table 2 on the page

Table 2. The parameters of liquid samples under study: the percentage of glycerol (β), dielectric constant (ϵl), density (ρl), specific conductivity (σl), shear viscosity measured by a viscometer (η66ll), shear elastic module (C66l), shear coefficient of viscosity (η66l), shear acoustic wave velocity (v66l), additional capacitance (Ca) and resistance (Ra) obtained as a result of fitting for liquids based on water and glycerol mixture with different viscosity and conductivity

Explanation and further investigation are required to connect the data in Figures 8 and 9. Is there any relationship with the QCM data?

Thank you for this comment. We have corrected the following text on the page 11.

Based on the data presented in the Table 2 the dependences of the additional capacity Ca on the conductivity of the liquid were constructed for various values of the volume concentration of glycerol. They are shown in Figure 8 (a). It can be seen that this capacity is practically independent on the conductivity of the liquid and decreases with increasing the glycerol concentration. The additional capacity Ca is an addition to the capacitance of the free resonator C0 due to the permittivity of the contacting liquid. With increasing the glycerol concentration, the permittivity of the mixture decreases from 80 to 57 and this leads to a decrease in the capacity Ca. This is confirmed by the Figure 8 (b), which shows the dependence of the average value of additional capacity <Ca> on the measured relative dielectric constant ε. Averaging was carried out for all conductivity values for each glycerol concentration. One can see that this capacity increases monotonically with increasing liquid permittivity. Therefore, the dependence presented in Figure 8 (b) may be used as a calibration curve to determine the liquid permittivity from the calculated value of the additional capacity.

Based on the data obtained, the dependences of the additional capacity Ca on the conductivity of the liquid were constructed for various values of the volume concentration of glycerol, see Figure 8 (a). It can be seen that this capacity is practically independent on the conductivity of the liquid and decreases with increasing the glycerol concentration. Figure 8 (b) shows the dependence of the average value of additional capacitance <Ca> on the measured relative dielectric constant ε. One can see that the capacitance increases monotonically with increasing dielectric constant.

We have corrected also the following text on the page 12.

It is obvious that when the resonator is completely immersed in liquid, the resonator electrodes are shunted by both an additional capacitance Ca (due to the dielectric constant) and an additional active resistance Ra (due to its conductivity). In the equivalent circuit, these elements are connected to the electrical input parallel to the resonator capacitance C0. Using the data given in Table 2, the dependence of the additional resistance Ra on the conductivity σ of the liquid was constructed. This dependence is shown in Figure 9. It can be seen that this resistance is practically independent of the volumetric content of glycerol (that is, on viscosity and permittivity) and monotonically decreases with increasing conductivity of the liquid. Therefore, this dependence can be used as a calibration curve to determine the conductivity σ of a liquid from the calculated value Ra.

Figure 9 shows the dependence of the additional resistance Ra on the conductivity of the liquid. It can be seen that this resistance is practically independent of the volumetric content of glycerol (that is, on viscosity) and decreases with increasing conductivity of the liquid.

The discussion stated that the additional resonator capacitance is weakly related to conductivity. To which resonator capacitance the liquid conductivity has little effect should be clear.

The free and loaded resonator in the electrical part has a capacitance C0 connected in parallel to the electrical input and an element –C0 connected in series with the electrical winding of the electromechanical transformer (see Fig. 6). When electrical voltage is applied to the electrical input of the resonator namely this element –C0 creates a driving mechanical force in the mechanical part of the circuit, which causes the resonator to oscillate. Upon contact with liquid, the additional capacitance and resistance are connected to the electrical part of the circuit, which depend on the dielectric constant and conductivity of the liquid, respectively. In general, a resonator circuit is obtained that takes into account the influence of the liquid. Information regarding additional capacity is provided in the previous answer.

We have corrected the text on the page 13:

It has been experimentally established The Table 2 shows that the additional capacitance of a resonator immersed in a conducting liquid weakly depends on the conductivity of the liquid.

Reviewer 3 Report

Comments and Suggestions for Authors

This paper studied the possibility of measuring the electrical and mechanical parameters of a liquid using a piezoelectric resonator with a longitudinal electric field. This paper is well organized and is solid with detailed description. I therefore recommend its publication in Sensors. Here are some issues to be clarified to improve the study.

1-      To clearly understand the device structure for readers, more information about the quartz resonator in this work should be provided. What’s the Euler angles of AT‐cut? What’s material for electrode?...

2-      when providing figures such as liquid conductivity, elastic modules, viscosity coefficients, are the authors sure that the precision of their figures is meaningful? Could the authors check their uncertainty range and provide values rounded accordingly?

3-      In Figure 7what causes a slight difference between the theory and experiment?

4-      Figure5 shows the frequency dependences of the electrical impedance of the AT‐quartz resonator. It seems that frequency dependence does not have enough measured points or frequency step. Will this affect the fitting accuracy?

Author Response

This paper studied the possibility of measuring the electrical and mechanical parameters of a liquid using a piezoelectric resonator with a longitudinal electric field. This paper is well organized and is solid with detailed description. I therefore recommend its publication in Sensors. Here are some issues to be clarified to improve the study.

The authors are grateful to Reviewer # 3 for the high assessment of our article.

  • To clearly understand the device structure for readers, more information about the quartz resonator in this work should be provided. What’s the Euler angles of ATcut? What’s material for electrode?...

We agree with this comment and have added this information on the page 4:

The selected resonator based on the quartz plate of AT or YXl/+350 – cut [33] (Euler angels = 0, 0, 350) for studying liquids had the plate thickness of 370 µm and electrode diameter of 5.8 mm. The electrode material is a complex silver-based alloy.

[33] Vig J. R., Quartz Crystal Resonators and Oscillators For Frequency Control and Timing Applications - A Tutorial, US Army Communications-Electronics Research, Development & Engineering Center Fort Monmouth, NJ, USA, 2004

  • when providing figures such as liquid conductivity, elastic modules, viscosity coefficients, are the authors sure that the precision of their figures is meaningful? Could the authors check their uncertainty range and provide values rounded accordingly?

We do not yet know how to check exactly the uncertainty range of the obtained values, but we have the following information. (1) The accuracy of measuring the resonator impedance by an impedance analyzer in our case is less than ±1%. (2) Many years of experience with equivalent circuits shows that each circuit parameter is also determined by the same error of ±1%. So we have rounded all the values presented in the tables to 2-3 significant numbers. We plan to study the problem of the accuracy of our method in details at the next stage of work.

  • In Figure 7what causes a slight difference between the theory and experiment?

The maximum difference between the theory and experiment is 6.8% for the real part of the impedance (7c) and 2.4% for the imaginary part of the impedance (7d). Taking into account the errors indicated in the previous answer, we can say that the maximum discrepancies are within reasonable limits. The main source of error is that the equivalent circuit method is developed in the approximation of plane waves propagating between the electrodes [34]. In reality, the aperture of these waves is limited by the size of the electrodes.

[34] Royer, D.; Dieulesaint, E, Elastic waves in solids II. Generation, acousto-optic interaction, applications. Springer-Verlag Berlin Heidelberg New York, 1999, ISBN 1439-2674

We have corrected the text on the page 9:

There is a good agreement between the theory and experiment and the maximum differences are equal 6.8% and 2.4% for the real and imaginary parts of the impedance, respectively.

  • Figure5 shows the frequency dependences of the electrical impedance of the ATquartz resonator. It seems that frequency dependence does not have enough measured points or frequency step. Will this affect the fitting accuracy?

We used a quartz resonator with a very high quality factor (Q = 1300000) (see the page 4). Therefore, there were relatively few points around the resonance. Naturally, this should affect the accuracy of the results. Nevertheless, the fitting of the theory to experiment for a free resonator turned out to be qualitative (Fig. 5). In the future, we plan to use resonators with a lower quality factor, since contact with liquid significantly reduces its value.

Round 2

Reviewer 2 Report

Comments and Suggestions for Authors

This paper presents the use of QCM sensors to detect several luqid parameters. With the additional correction based on the comment, the results and discussion have been presented well and clearly. This article provides insight into the possible uses of QCM sensors and also provides a good perspective for further research and development by the author and readers.